# LMI-Based MPC Design Applied to the Single-Phase PWM Inverter with LC Filter under Uncertain Parameters

**DOI:** 10.3390/s24134325

**Published:** 2024-07-03

**Authors:** Cristiano Quevedo Andrea, Edson Antonio Batista, Luís Felipe da Silva Carlos Pereira, Moacyr Aureliano Gomes de Brito, Gustavo Vargas de Souza

**Affiliations:** 1Faculty of Engineering, Architecture and Urban Planning and Geography, Federal University of Mato Grosso do Sul, Campo Grande 79070-900, MS, Brazil; edson.batista@ufms.br (E.A.B.); luis.pereira@ufms.br (L.F.d.S.C.P.); moacyr.brito@ufms.br (M.A.G.d.B.); 2Department of Electrical Engineering, School of Engineering, São Paulo State University, Ilha Solteira 15385-000, SP, Brazil; gustavo.vargas@unesp.br

**Keywords:** single-phase PWM inverter, FPGA-in-the-loop, predictive control, linear matrix inequalities

## Abstract

This work proposes a design methodology for predictive control applied to the single-phase PWM inverter with an LC filter. In the design, we considered that the PWM inverter has parametric uncertainties in the filter inductance and output load resistance. The control system purpose is to track a sinusoidal signal at the inverter output. The designed control system with an embedded integrator uses the principle of receding horizon control, which underpinned predictive control. The methodology was described by linear matrix inequalities, which can be solved efficiently using convex programming techniques, and the optimal solution is obtained. MATLAB-Simulink and real-time FPGA-in-the-loop simulations illustrate the viability of the proposed control system. The LMI-based MPC reveals an effective performance for tracking of a sinusoidal reference signal and disturbance rejection of input voltage and load perturbations for the inverter subject to uncertainties.

## 1. Introduction

DC-AC converters, also known as inverters, are power electronics systems that perform power level conversion. These converters can convert continuous levels of voltage or current on alternate levels green at their output, with symmetry and frequency range specified in the project [1,2,3]. Many residential and industrial applications use these devices [4,5], and can be classified as power supplies that function as voltage or current sources. In addition, the single-phase topology implementation of DC-AC inverters can be push-pull, half-bridge, or full-bridge types.

In an uninterruptible power supply (UPS), the inverter makes up the output stage of this device and works as a voltage source [6]. The fundamental principle of UPS is to provide a controlled sinusoidal voltage with precise desired specifications. Then, an LC filter is used to remove the harmonics produced during the high-frequency switching of the semiconductor switches. In operation, it is standard for inverters to be subject to parametric variations and external disturbances, so the control system must provide stability and robustness as well as voltage regulation. Research addressing control applied to inverters has been modernizing over time. Initially, Ref. [7] introduced a controller employing load-current derivative feedback to reject disturbances at the output of single-phase UPS inverters. During the same period, we also observed the application of predictive control, implemented in a DSP, aimed at reducing tracking error amplitudes and control signal variations [8].

To design a tracking system for PWM inverters, one can employ repetitive control. In Ref. [9], researchers presented a control system that comprises state feedback and repetitive control for output regulation of PWM inverters. The control system suggested in Ref. [9] is tuned to the frequency of the reference signal, and for plants with un-modeled delays, a repetitive control compensation filter is determined by experiments. A model-predictive control (MPC) was proposed in Ref. [10] to provide a fast response with the control of PWM inverters connected to the grid through resonant LCL filters without uncertain parameters.

A Field Programmable Gate Array (FPGA) can be used to implement control systems [11,12], including MPC. For instance, in Ref. [13], an FPGA was utilized to implement an MPC control strategy for three-phase four-leg grid-tied inverters. This implementation incorporated tracking error in the cost function as a criterion for optimizing the duty cycle of the inverters. In Ref. [14], an FPGA was used in order to embed a fixed-switching-frequency modulated model predictive control strategy as an inner controller for a two-level, three-phase voltage source inverter operating within an islanded AC microgrid.

A comparison between predictive control and classical PI control applied to a single-phase inverter with output LC filter can be seen in Ref. [15]. The authors proposed a control algorithm that maintains the output voltage at the desired reference with any kind of load and minimizes the switching frequency. The results of the predictive control system were satisfactory when compared with the PI regulators and PWM. In Ref. [16], the authors use a predictive and repetitive control technique to regulate the operation of the grid-connected inverter under distorted voltage conditions. The controllers were designed using the modeling by state-space in stationary reference frame (αβ). The methodology demonstrated good performance in mitigating harmonics in cases where the network voltage is distorted. However, the increase in the prediction horizon led to higher computational burden.

A control system employing a finite control set model predictive control and load current estimation applied to the single-phase inverter with LC filter was introduced in Ref. [17]. The control strategy was set up with the Speedgoat real-time target machine and presented suitable regulation to the voltage output of the converter. Nonetheless, the methodology proposed in Ref. [17] did not address uncertainties in the inverter model. Furthermore, one can also observe the application of adaptive control techniques in inverters. Ref. [18] presented an observer-based adaptive control scheme for single-phase UPS inverters under nonlinear load. The stability of the system is analyzed by the Lyapunov approach and it was used a state observer to estimate the tracking error vector in order to obtain output-voltage tracking.

In Ref. [19] the authors presented a design of a control system by linear matrix inequalities that provides robustness to uncertainties on grid inductance for grid-connected PWM inverter with LCL filters. Control system design via LMI has been used in various areas for engineering problem solutions [20,21,22,23]. Since controller design is formulated in terms of LMIs, closed-loop stability is guaranteed, and it is possible to insert control design constraints such as bound for control input and output and polytopic uncertainties [24]. A disturbance rejection system based on mixed $H_{2}/H_{\infty }$ can be found in Ref. [25]. The authors considered uncertainties in the inverter model and used model predictive in order to control the inverters that were connected in parallel to renewable energy sources. In the case of uncertain systems or systems modeled with actuator failures, robust controllers with guaranteed costs are employed. In Ref. [26], a robust and reliable guaranteed cost for a fuzzy controller applied to discrete-time nonlinear systems is proposed. The system addressed in the work involves time-varying delays, and the controller was designed using LMIs.

In this work, we proposed the design of a predictive control system applied to a single-phase full-bridge PWM inverter with resistive and resistive-inductive loads to achieve regulated output sinusoidal voltage. This control system uses the feedback of the predicted states with integral action, and the project is formulated in terms of LMIs, considering parametric uncertainties in the inverter model. In this case, we considered that the inverter presents parametric uncertainties in filter inductance and load resistance. The design of the proposed LMI-MPC was carried out in the discrete-time domain and its implementation developed in FPGA. Simulation in FPGA-in-the-loop (FIL) based on MATLAB/Simulink environment as performed in Ref. [27] demonstrates that the proposed control system provides suitable solution for sinusoidal-reference tracking and disturbance rejection for load and input voltage perturbations to the inverter with polytopic uncertainties.

Following Section 1, this work is organized as follows: Section 2 provides the modeling of full-bridge PWM inverter state-space, and Section 3 describes predictive control with integral action for an uncertain system based on LMIs. The performance comparison between the ACMC and the proposed LMI-MPC is presented in Section 5. In addition, Section 6 presents the results of the FIL simulation applied to the inverter, and the conclusions are drawn in Section 8.

## 2. Full-Bridge PWM Inverter State-Space Modelling

Considering a full-bridge PWM inverter illustrated in Figure 1, and using Kirchhoff’s laws, one can obtain the differential equations that represent the inverter with the LC filter. In the case of the inverter with a resistive load ($Z_{c}=R_{c}$) we have the following differential equations [28],
(1)$$-u\left(t\right)+L_{c1}\frac{di_{L1}\left(t\right)}{dt}+v_{c}\left(t\right)=0,$$
(2)$$i_{L1}\left(t\right)-C_{c}\frac{dv_{c}\left(t\right)}{dt}-\frac{v_{c}\left(t\right)}{R_{c}}=0.$$

By describing (Equation 1) and (Equation 2) into matrices, one can obtain the state-space representation for the system illustrated in Figure 1. Thus, one has,
(3)$$\begin{array}{cc} \left[\begin{array}{c} {\dot{x}}_{1}\left(t\right)\\ {\dot{x}}_{2}\left(t\right) \end{array}\right]\; & =\overset{A_{R}}{\overbrace{\left[\begin{array}{cc} -\frac{1}{R_{c}C_{c}} & \frac{1}{C_{c}}\;\\ -\frac{1}{L_{c1}} & 0 \end{array}\right]}}\overset{x(t)}{\overbrace{\left[\begin{array}{c} x_{1}\left(t\right)\\ x_{2}\left(t\right) \end{array}\right]}}+\overset{B_{R}}{\overbrace{\left[\begin{array}{c} 0\\ \frac{1}{L_{c1}} \end{array}\right]}}u\left(t\right), \end{array}$$
(4)$$\begin{array}{cc} y(t) & =\overset{C_{R}}{\overbrace{\left[\begin{array}{cc} 1 & 0 \end{array}\right]}}\left[\begin{array}{c} x_{1}\left(t\right)\\ x_{2}\left(t\right) \end{array}\right], \end{array}$$
where $x_{1}\left(t\right)=v_{c}\left(t\right)$, $x_{2}\left(t\right)=i_{L1}\left(t\right)$, u(t) is the control input and y(t) is the output of the system. According to (Equation 4), the system output is the capacitor voltage, as shown in Figure 1.

On the other hand, if the inverter presents a resistive-inductive load ($Z_{c}=R_{c}+j\omega L_{c2}$), one has the following differential equations to describe the system: (5)$$\begin{array}{cc} -u\left(t\right)+L_{c1}\frac{di_{L1}\left(t\right)}{dt}+v_{c}\left(t\right) & =0, \end{array}$$(6)$$\begin{array}{cc} L_{c2}\frac{di_{L2}\left(t\right)}{dt}+R_{c}i_{L2}\left(t\right)-v_{c}\left(t\right) & =0, \end{array}$$(7)$$\begin{array}{cc} i_{L1}\left(t\right)-C\frac{dv_{c}\left(t\right)}{dt}-i_{L2}\left(t\right) & =0. \end{array}$$

The final result for the state-space representation of the inverter with resistive-inductive load is found by (Equation 5)–(Equation 7) in vector-matrix form as follows [29],
(8)$$\begin{array}{cc} \left[\begin{array}{c} {\dot{x}}_{1}\left(t\right)\\ {\dot{x}}_{2}\left(t\right)\\ {\dot{x}}_{3}\left(t\right) \end{array}\right]\; & =\overset{A_{RL}}{\overbrace{\left[\begin{array}{ccc} 0 & \frac{1}{C_{c}} & -\frac{1}{C_{c}}\;\\ -\frac{1}{L_{c1}} & 0 & 0\\ \frac{1}{L_{c2}} & 0 & -\frac{R_{c}}{L_{c2}} \end{array}\right]}}\overset{x(t)}{\overbrace{\left[\begin{array}{c} x_{1}\left(t\right)\\ x_{2}\left(t\right)\\ x_{3}\left(t\right) \end{array}\right]}}+\overset{B_{RL}}{\overbrace{\left[\begin{array}{c} 0\\ \frac{1}{L_{c1}}\\ 0 \end{array}\right]}}u\left(t\right), \end{array}$$
(9)$$\begin{array}{cc} y(t) & =\overset{C_{RL}}{\overbrace{\left[\begin{array}{ccc} 1 & 0 & 0 \end{array}\right]}}\left[\begin{array}{c} x_{1}\left(t\right)\\ x_{2}\left(t\right)\\ x_{3}\left(t\right) \end{array}\right], \end{array}$$
where $x_{3}\left(t\right)=i_{L2}\left(t\right)$.

In (Equation 3) and (Equation 8) we considered that
(10)$$\begin{array}{c} R_{c_{\min}}\leq R_{c}\leq R_{c_{\max}}\;and\;L_{c_{\min}}\leq L_{c}\leq L_{c_{\max}}. \end{array}$$

The systems described in (Equation 3) and (Equation 8) become uncertain when the condition (Equation 10) is considered. In (Equation 11), we present the state-space model for the PWM inverter, which incorporates parametric uncertainties.
(11)$$\begin{array}{cc} \dot{x}\left(t\right) & =A(\alpha )x(t)+B(\alpha )u(t),\\ y(t) & =C(\alpha )x(t). \end{array}$$

The uncertain dynamic matrices A(α), B(α) and C(α) are represented by the convex combination described below [30],
(12)$$\begin{array}{c} A\left(\alpha \right)=\sum_{i=1}^{p}\alpha_{i}A_{i},\;B\left(\alpha \right)=\sum_{i=1}^{p}\alpha_{i}B_{i}\;and\;C\left(\alpha \right)=\sum_{i=1}^{p}\alpha_{i}C_{i}, \end{array}$$
where $\alpha ={\left[\begin{array}{ccc} \alpha_{1} & \cdots & \alpha_{p} \end{array}\right]}^{\prime }$ is the vector that parameterizes the polytope uncertainties, and still,
(13)$$\begin{array}{c} \sum_{i=1}^{p}\alpha_{i}=1,\;\alpha_{i}\geq 0. \end{array}$$

The number of vertices of the uncertainties polytope varies from 1 to *p*, where $p=2^{v}$ and *v* represent the number of uncertain parameters of the plant.

In the predictive control system design applied to the inverter that considers resistive loads, we have the system $\left(A_{R}\left(\alpha \right),B_{R}\left(\alpha \right),C_{R}\left(\alpha \right)\right)=\left(A\left(\alpha \right),B\left(\alpha \right),C\left(\alpha \right)\right)$. In the case of resistive-inductive loads, one has the system $\left(A_{RL}\left(\alpha \right),B_{RL}\left(\alpha \right),C_{RL}\left(\alpha \right)\right)=\left(A\left(\alpha \right),B\left(\alpha \right),C\left(\alpha \right)\right)$. $S_{R}=\left(A_{R}\left(\alpha \right),B_{R}\left(\alpha \right),C_{R}\left(\alpha \right)\right)$ and $S_{RL}=\left(A_{RL}\left(\alpha \right),B_{RL}\left(\alpha \right),C_{RL}\left(\alpha \right)\right)$ are the uncertain models of the systems given in (Equation 3) and (Equation 8), respectively.

In this manuscript, we aim to solve the problem of the sinusoidal voltage regulation of the PWM inverter with polytope uncertain, which can be summarized as follows:

**Problem** **1.**
*The problem of sinusoidal voltage regulation of the PWM inverter, considering parametric uncertainties in the inverter model given in (Equation 11), is to design a discrete predictive controller with state feedback and integral action, ensuring asymptotic stability for the closed-loop system.*


The block diagrams of the control system used to solve Problem 1 in this manuscript is provided in Figure 2. In this diagram, one can see the LMI-MPC controller $K=\left[\begin{array}{cc} K_{x} & K_{y} \end{array}\right]$, which has the function of regulating the output voltage level of the inverter, providing tracking of the sinusoidal reference signal.

## 3. Predictive Model with Integral Action

Consider the following discrete state-space model for the uncertain system given in (Equation 11) as
(14)$$\begin{array}{cc} x(k+1) & =A_{d}\left(\alpha \right)x\left(k\right)+B_{d}\left(\alpha \right)u\left(k\right),\\ y(k) & =C_{d}\left(\alpha \right)x\left(k\right), \end{array}$$
where $x\left(k\right)\in R^{n_{1}\times 1}$, $A_{d}\left(\alpha \right)\in R^{n_{1}\times n_{1}}$, $B_{d}\left(\alpha \right)\in R^{n_{1}\times 1}$, u∈R, $C_{d}\left(\alpha \right)\in R^{1\times n_{1}}$ and y(k)∈R. In other words, the plant has 1 input, 1 output, and $n_{1}$ states. To insert the integrator into the predictive control system [31], an integral signal must be applied to the plant input. This effect can be achieved by considering that the input control is u(k)−u(k−1). So subtracting both sides of (Equation 14) from the variables in their previous state results in,
(15)$$\begin{array}{c} x\left(k+1\right)-x\left(k\right)=A_{d}\left(\alpha \right)\left(x(k)-x(k-1)\right)+B_{d}\left(\alpha \right)\left(u(k)-u(k-1)\right). \end{array}$$

The differences between the state variables and control law considering the actual instant and the previous one can be calculated as,
(16)$$\begin{array}{c} \Delta x(k+1)=x(k+1)-x(k),\Delta x(k)=x(k)-x(k-1),\Delta u(k)=u(k)-u(k-1). \end{array}$$

With this transformation, one can write the state-equation representation of the system as,
(17)$$\begin{array}{c} \Delta x\left(k+1\right)=A_{d}\left(\alpha \right)\Delta x\left(k\right)+B_{d}\left(\alpha \right)\Delta u\left(k\right). \end{array}$$

Similarly, for the output signal given in (Equation 14), subtract y(k+1) from y(k), we obtain,
(18)$$\begin{array}{cc} y(k+1)-y(k) & =C_{d}\left(\alpha \right)\left(x(k+1)-x(k)\right),\\ \; & =C_{d}\left(\alpha \right)\Delta x\left(k+1\right). \end{array}$$
which results in,
(19)$$\begin{array}{c} \Delta y\left(k+1\right)=C_{d}\left(\alpha \right)A_{d}\left(\alpha \right)\Delta x\left(k\right)+C_{d}\left(\alpha \right)B_{d}\left(\alpha \right)\Delta u\left(k\right). \end{array}$$

Choosing,
(20)$$\begin{array}{c} x_{e}\left(k\right)=\left[\begin{array}{c} \Delta x(k)\\ y(k) \end{array}\right], \end{array}$$
one can organize Equations (Equation 17) and (Equation 19) in the matrix form to obtain the discrete state-space representation with an embedded integrator as [32],
(21)$$\begin{array}{cc} \left[\begin{array}{c} \Delta x(k+1)\\ \Delta y(k+1) \end{array}\right]\; & =\overset{A_{e}\left(\alpha \right)}{\overbrace{\left[\begin{array}{cc} A_{d}\left(\alpha \right) & 0_{d}^{\prime }\\ C_{d}\left(\alpha \right)A_{d}\left(\alpha \right) & 1 \end{array}\right]}}\overset{x_{e}\left(k\right)}{\overbrace{\left[\begin{array}{c} \Delta x(k)\\ y(k) \end{array}\right]}}+\overset{B_{e}\left(\alpha \right)}{\overbrace{\left[\begin{array}{c} B_{d}\left(\alpha \right)\\ C_{d}\left(\alpha \right)B_{d}\left(\alpha \right) \end{array}\right]}}\Delta u\left(k\right),\\ y(k) & =\overset{C_{e}\left(\alpha \right)}{\overbrace{\left[\begin{array}{cc} 0_{d} & 1 \end{array}\right]}}\left[\begin{array}{c} \Delta x(k)\\ y(k)1 \end{array}\right]. \end{array}$$
where $0_{d}=\overset{n_{1}}{\overbrace{\left[\begin{array}{cccc} 0 & 0 & \cdots & 0 \end{array}\right]}}$.

Equation (Equation 21), with $u_{e}\left(k\right)=\Delta u\left(k\right)$, can still be written in the form,
(22)$$\begin{array}{cc} x_{e}\left(k+1\right) & =A_{e}\left(\alpha \right)x_{e}\left(k\right)+B_{e}\left(\alpha \right)u_{e}\left(k\right),\\ y(k) & =C_{e}\left(\alpha \right)x_{e}\left(k\right). \end{array}$$

The model obtained in (Equation 21) is referred to as the augmented system model [33] and it was used in the predictive controller design proposed in this work. The control based on model predictive is a closed-loop control that uses a plant model to make predictions about the behavior of its output. For this, it also uses an optimizer, which ensures that the predicted output tracks the desired reference.

At the sampling instant $k_{i}$, where $k_{i}>0$, consider the state vector $x_{e}\left(k_{i}\right)$ as available, and take the states of $x_{e}\left(k_{i}\right)$ as the current augmented state of the plant. The future trajectory of the control signal is denoted by,
(23)$$u_{e}\left(k_{i}\right),\;u_{e}\left(k_{i}+1\right),\;\ldots ,\;u_{e}\left(k_{i}+N_{c}-1\right),$$
where $N_{c}$ is the control horizon. The horizon of control is a parameter used to capture the future trajectory of the control signal.

The future state variables of $x_{e}\left(k_{i}\right)$ are predicted by $N_{p}$ samples, where $N_{p}$ is the prediction horizon, and it can be described as,
(24)$$\begin{array}{c} x_{e}\left(k_{i}+1|k_{i}\right),\;x_{e}\left(k_{i}+2|k_{i}\right),\;\ldots ,\;x_{e}\left(k_{i}+m|k_{i}\right),\;\cdots ,\;x_{e}\left(k_{i}+N_{p}|k_{i}\right). \end{array}$$

The variable $x_{e}\left(k_{i}+m|k_{i}\right)$ is the prediction of the state variable at $k_{i}+m$ given $x_{e}\left(k_{i}\right)$. The control horizon is always less than or equal to the prediction horizon. Thus, considering (Equation 22), the state-space representation for future variables can be obtained as,
(25)$$\begin{array}{cc} x_{e}\left(k_{i}+1|k_{i}\right) & =A_{e}\left(\alpha \right)x_{e}\left(k_{i}\right)+B_{e}\left(\alpha \right)u_{e}\left(k_{i}\right),\\ x_{e}\left(k_{i}+2|k_{i}\right) & =A_{e}{\left(\alpha \right)}^{2}x_{e}\left(k_{i}\right)+A_{e}\left(\alpha \right)B_{e}\left(\alpha \right)u_{e}\left(k_{i}\right)+B_{e}\left(\alpha \right)u_{e}\left(k_{i}+1\right),\\ \; & \vdots \\ x_{e}\left(k_{i}+N_{p}\;|k_{i}\right) & =A_{e}{\left(\alpha \right)}^{N_{p}}x_{e}\left(k_{i}\right)+A_{e}{\left(\alpha \right)}^{N_{p}-1}B_{e}\left(\alpha \right)u_{e}\left(k_{i}\right)+A_{e}{\left(\alpha \right)}^{N_{p}-2}B_{e}\left(\alpha \right)u_{e}\left(k_{i}+1\right)+\\ \; & \cdots +A_{e}{\left(\alpha \right)}^{N_{p}-N_{c}}B_{e}\left(\alpha \right)u_{e}\left(k_{i}+N_{c}-1\right). \end{array}$$

Using the predicted state variables, it is possible to obtain the predicted outputs as follows:(26)$$\begin{array}{cc} y_{e}\left(k_{i}+1|k_{i}\right) & =C_{e}\left(\alpha \right)A_{e}\left(\alpha \right)x_{e}\left(k_{i}\right)+C_{e}\left(\alpha \right)B_{e}\left(\alpha \right)u_{e}\left(k_{i}\right),\\ y_{e}\left(k_{i}+2|k_{i}\right) & =C_{e}\left(\alpha \right)A_{e}{\left(\alpha \right)}^{2}x_{e}\left(k_{i}\right)+C_{e}\left(\alpha \right)A_{e}\left(\alpha \right)B_{e}\left(\alpha \right)u_{e}\left(k_{i}\right)+C_{e}\left(\alpha \right)B_{e}\left(\alpha \right)u_{e}\left(k_{i}+1\right),\\ y_{e}\left(k_{i}+3|k_{i}\right) & =C_{e}\left(\alpha \right)A_{e}{\left(\alpha \right)}^{3}x_{e}\left(k_{i}\right)+C_{e}\left(\alpha \right)A_{e}{\left(\alpha \right)}^{2}B_{e}\left(\alpha \right)u_{e}\left(k_{i}\right)+\\ \; & C_{e}\left(\alpha \right)A_{e}\left(\alpha \right)B_{e}\left(\alpha \right)u_{e}\left(k_{i}+1\right)+C_{e}\left(\alpha \right)B_{e}\left(\alpha \right)u_{e}\left(k_{i}+2\right),\\ \; & \vdots \\ y_{e}\left(k_{i}+N_{p}|k_{i}\right) & =C_{e}\left(\alpha \right)A_{e}{\left(\alpha \right)}^{N_{p}}x_{e}\left(k_{i}\right)+C_{e}\left(\alpha \right)A_{e}{\left(\alpha \right)}^{N_{p}-1}B_{e}\left(\alpha \right)u_{e}\left(k_{i}\right)+\\ \; & C_{e}\left(\alpha \right)A_{e}{\left(\alpha \right)}^{N_{p}-2}B_{e}\left(\alpha \right)u_{e}\left(k_{i}+1\right)+\\ \; & \cdots +C_{e}\left(\alpha \right)A_{e}{\left(\alpha \right)}^{N_{p}-N_{c}}B_{e}\left(\alpha \right)u_{e}\left(k_{i}+N_{c}-1\right). \end{array}$$

Note that all predicted variables are a function of the current state variable, $x_{e}\left(k_{i}\right)$, and the future control action, $u_{e}\left(k_{i}+n\right)$, where $n=0,1,\ldots \left(N_{c}-1\right)$. Then, it is possible to write the following vectors: (27)$$\begin{array}{cc} X(k_{i}+1) & ={\left[x_{e}\left(k_{i}+1|k_{i}\right)\;x_{e}\left(k_{i}+2|k_{i}\right)\;x_{e}\left(k_{i}+3|k_{i}\right)\;\ldots x_{e}\left(k_{i}+N_{p}|k_{i}\right)\right]}^{\prime }, \end{array}$$(28)$$\begin{array}{cc} Y(k_{i}+1) & ={\left[y_{e}\left(k_{i}+1|k_{i}\right)\;y_{e}\left(k_{i}+2|k_{i}\right)\;y_{e}\left(k_{i}+3|k_{i}\right)\;\ldots y_{e}\left(k_{i}+N_{p}|k_{i}\right)\right]}^{\prime }, \end{array}$$(29)$$\begin{array}{cc} U(k_{i}) & ={\left[u_{e}\left(k_{i}\right)\;u_{e}\left(k_{i}+1\right)\;u_{e}\left(k_{i}+2\right)\;u_{e}\left(k_{i}+3\right)\;\ldots \;u_{e}\left(k_{i}+N_{c}-1\right)\right]}^{\prime }. \end{array}$$

Therefore, a model predictive can be written based on the vectors (Equation 27)–(Equation 29) and on the Equations (Equation 25) and (Equation 26) as follows,
(30)$$\begin{array}{cc} X(k_{i}+1) & =G\left(\alpha \right)X\left(k_{i}\right)+\Gamma \left(\alpha \right)U\left(k_{i}\right), \end{array}$$
(31)$$\begin{array}{cc} Y(k_{i}+1) & =F\left(\alpha \right)X\left(k_{i}\right)+\Phi \left(\alpha \right)U\left(k_{i}\right). \end{array}$$

The predictive matrices Φ(α), Γ(α), F(α) and G(α) are shown in (Equation 32).
(32)$$\begin{array}{cc} \Phi (\alpha ) & =\left[\begin{array}{cccc} C_{e}\left(\alpha \right)B_{e}\left(\alpha \right) & 0 & \ldots & 0\\ C_{e}\left(\alpha \right)A_{e}\left(\alpha \right)B_{e}\left(\alpha \right) & C_{e}\left(\alpha \right)B_{e}\left(\alpha \right) & \ldots & 0\\ \vdots & \vdots & \ldots & \vdots \\ C_{e}\left(\alpha \right)A_{e}{\left(\alpha \right)}^{N_{p}-1}B_{e}\left(\alpha \right) & C_{e}\left(\alpha \right)A_{e}{\left(\alpha \right)}^{N_{p}-2}B_{e}\left(\alpha \right)\; & \ldots & C_{e}\left(\alpha \right)A_{e}{\left(\alpha \right)}^{N_{p}-N_{c}}B_{e}\left(\alpha \right) \end{array}\right],\\ \Gamma (\alpha ) & =\left[\begin{array}{cccc} B_{e}\left(\alpha \right) & 0 & \cdots & 0\\ A_{e}\left(\alpha \right)B_{e}\left(\alpha \right) & B_{e}\left(\alpha \right) & \ldots & 0\\ \vdots & \vdots & \cdots & \vdots \\ A_{e}{\left(\alpha \right)}^{Np-1}B_{e}\left(\alpha \right) & A_{e}{\left(\alpha \right)}^{Np-2}B_{e}\left(\alpha \right) & \ldots & A_{e}{\left(\alpha \right)}^{Np-Nc}B_{e}\left(\alpha \right) \end{array}\right],\\ F(\alpha ) & =\left[\begin{array}{cc} C_{e}\left(\alpha \right)A_{e}\left(\alpha \right) & 0_{1\times \left(N_{p}-1\right)*\left(n_{1}+1\right)}\\ C_{e}\left(\alpha \right)A_{e}{\left(\alpha \right)}^{2} & 0_{1\times \left(N_{p}-1\right)*\left(n_{1}+1\right)}\\ C_{e}\left(\alpha \right)A_{e}{\left(\alpha \right)}^{3} & 0_{1\times \left(N_{p}-1\right)*\left(n_{1}+1\right)}\\ \vdots & \vdots \\ C_{e}\left(\alpha \right)A_{e}{\left(\alpha \right)}^{N_{p}} & 0_{1\times \left(N_{p}-1\right)*\left(n_{1}+1\right)} \end{array}\right],G\left(\alpha \right)=\left[\begin{array}{cc} A_{e}\left(\alpha \right) & 0_{\left(n_{1}+1\right)\times \left(N_{p}-1\right)*\left(n_{1}+1\right)}\\ A_{e}{\left(\alpha \right)}^{2} & 0_{\left(n_{1}+1\right)\times \left(N_{p}-1\right)*\left(n_{1}+1\right)}\\ A_{e}{\left(\alpha \right)}^{3} & 0_{\left(n_{1}+1\right)\times \left(N_{p}-1\right)*\left(n_{1}+1\right)}\\ \vdots & \vdots \\ A_{e}{\left(\alpha \right)}^{N_{p}} & 0_{\left(n_{1}+1\right)\times \left(N_{p}-1\right)*\left(n_{1}+1\right)} \end{array}\right]. \end{array}$$

Here, G(α) must be a quadratic matrix to fulfill the LMI description for a feasible solution and for the model obtained in (Equation 31) to be valid. F(α) must follow the description provided in (Equation 32).

## 4. LMI-MPC Design

To design a predictive controller, one must seek a solution in which the predicted output tracks as closely as possible the desired reference and, simultaneously, to acquire control actions that avoid aggressive control manoeuvres as well as present a fast response. The way to achieve this balanced performance is by using a cost function and then, assigning weights to determine the importance level of the aforementioned requirements. For this purpose, the cost function to optimize the control action is based on the predicted augmented state vector (Equation 27) and the predicted augmented control vector (Equation 29).

Thus, given a reference $r(k_{i})$ at time $k_{i}$, it was considered that the vector containing the reference information is,
(33)$$\begin{array}{c} R=\overset{N_{p}}{\overbrace{\left[\begin{array}{cccc} 1 & 1 & \cdots & 1 \end{array}\right]}}\prime r\left(k_{i}\right)=R_{s}r\left(k_{i}\right), \end{array}$$
and the performance index is the cost function given by
(34)$$\begin{array}{c} J=\sum_{k_{i}=0}^{\infty }\left[U{\left(k_{i}\right)}^{\prime }R_{w}U\left(k_{i}\right)+X{\left(k_{i}\right)}^{\prime }QX\left(k_{i}\right)\right], \end{array}$$
where $R_{w}>0\in R^{N_{c}\times N_{c}}\;\mathrm{and}\;Q=Q^{\prime }>0\in R^{N_{p}\left(n_{1}+1\right)\times N_{p}\left(n_{1}+1\right)}$ are given weighing matrices.

The design goal of the proposed predictive control is as follows:To select a gain matrix *K* applying the control law $U\left(k_{i}\right)=-KX\left(k_{i}\right)$ such that the model predictive is asymptotically stable and the performance index (Equation 34) satisfies the upper bound,
(35)$$J\leq X{\left(0\right)}^{\prime }PX\left(0\right),$$
where *P* is a positive definite matrix.Minimize the upper bound $X{\left(0\right)}^{\prime }PX\left(0\right)$.

In order to derive the LMI conditions required for designing the predictive control system, we introduce Definition 1 and Lemma 1.

**Definition** **1.**
*The uncertain system $x_{c}\left(k+1\right)=A_{c}\left(\alpha \right)x\left(k\right)+B_{c}\left(\alpha \right)u_{c}\left(k\right)$, with $u_{c}\left(k\right)=-K_{c}x_{c}\left(k\right)$, considering all potential systems within the defined set of uncertainties for a specific value of α, is D-stable with all complex poles of the closed-loop within a region D illustrated in Figure 3, if and only if there exists a positive symmetric matrix P such that [34,35].*

(36)
$$\begin{array}{c} \left[\begin{array}{cc} -\rho P & A_{ci}P+qP\\ PA_{ci}^{\prime }+qP & -\rho P \end{array}\right]<0, \end{array}$$


*i=1,2,⋯,p.*


**Lemma** **1**(Schur complement). *For any symmetric matrix, W, of the form [36]*
(37)$$W=\left[\begin{array}{cc} Z_{1} & Z_{2}\\ Z_{2}^{\prime } & Z_{3} \end{array}\right],$$
*if $Z_{1}$ is invertible then the following properties hold:*
*$W>0\;\mathrm{iff}\;Z_{1}>0\;\mathrm{and}\;Z_{3}-Z_{2}^{\prime }Z_{1}^{-1}Z_{2}>0$.**$Z_{1}>0\;\mathrm{iff}\;W\geq 0\;\mathrm{and}\;Z_{3}-Z_{2}^{\prime }Z_{1}^{-1}Z_{2}\geq 0$.*

Theorems 1 and 2 were proposed to solve Problem 1 in order to design the LMI-MPC controller *K* described in Figure 2. In this sense, both theorems can be used for predictive controller design.

**Theorem** **1.**
*There exist guaranteed cost controllers that are D-stable for the uncertain system (Equation 31) and the cost function (Equation 34) if there exists a symmetric definite positive matrix M>0 and a matrix N such that,*

(38)
$$\begin{array}{cc} \left[\begin{array}{cccc} Q & 0 & 0 & QM\\ 0 & R_{w} & 0 & R_{w}N\\ 0 & 0 & M & G_{i}M-\Gamma_{i}N\\ MQ & N^{\prime }R_{w} & MG_{i}^{\prime }-N^{\prime }\Gamma_{i}^{\prime } & M \end{array}\right] & >0, \end{array}$$


(39)
$$\begin{array}{cc} \left[\begin{array}{cc} -\rho M & G_{i}M-\Gamma_{i}N+qM\\ MG_{i}^{\prime }-N^{\prime }\Gamma_{i}^{\prime }+qM & -\rho M \end{array}\right] & <0, \end{array}$$


*i=1,2,⋯,p.*

*Furthermore, if (M,N) is a feasible solution to the inequality (Equation 38), then $U\left(k_{i}\right)=-KX\left(k_{i}\right)$ is a guaranteed cost control law of system (Equation 31), where the feedback gain matrix K is given by*

(40)
$$K=NM^{-1}$$

*and the corresponding closed-loop cost function satisfies*

(41)
$$J\leq X{\left(0\right)}^{\prime }M^{-1}X\left(0\right).$$



**Proof.** Applying $U\left(k_{i}\right)=-KX\left(k_{i}\right)$ in (Equation 31), one obtains,
(42)$$\begin{array}{cc} X(k_{i}+1) & =\left(G\left(\alpha \right)-\Gamma \left(\alpha \right)K\right)X\left(k_{i}\right). \end{array}$$Then, consider the following Lyapunov candidate function for the system (Equation 31),
(43)$$V\left(X\left(k_{i}\right)\right)=X{\left(k_{i}\right)}^{\prime }PX\left(k_{i}\right).$$If $U(k_{i})$ is the control law of guaranteed cost for the system described in (Equation 31), the increment of $V(x\left(k_{i}\right))$ along the trajectory of the predictive system, according to Ref. [36], satisfies the following condition:
(44)$$\begin{array}{cc} \Delta V(X\left(k_{i}\right)) & =V\left(X\left(k_{i}+1\right)\right)-V\left(X\left(k_{i}\right)\right)\\ \; & =X{\left(k_{i}\right)}^{\prime }\left[{\left(G(\alpha )-\Gamma (\alpha )K\right)}^{\prime }P\left(G(\alpha )-\Gamma (\alpha )K\right)-P+K^{\prime }R_{w}K+Q\right]X\left(k_{i}\right)<0. \end{array}$$Then, provided that
(45)$$\begin{array}{c} {\left(G(\alpha )-\Gamma (\alpha )K\right)}^{\prime }P\left(G(\alpha )-\Gamma (\alpha )K\right)-P+K^{\prime }R_{w}K+Q<0,\\ \sum_{i=1}^{p}\alpha_{i}\left({\left(G_{i}-\Gamma_{i}K\right)}^{\prime }P\left(G_{i}-\Gamma_{i}K\right)-P+K^{\prime }R_{w}K+Q\right)<0, \end{array}$$
(46)$$\begin{array}{c} {\left(G_{i}-\Gamma_{i}K\right)}^{\prime }P\left(G_{i}-\Gamma_{i}K\right)-P+K^{\prime }R_{w}K+Q<0, \end{array}$$
holds for all potential systems within the defined set of uncertainties, the system described in (Equation 31), operating in closed-loop with $U\left(k_{i}\right)=-KX\left(k_{i}\right)$, is asymptotically stable [37].From (Equation 44), one has
(47)$$\begin{array}{c} X{\left(k_{i}\right)}^{\prime }\left[{\left(G(\alpha )-\Gamma (\alpha )K\right)}^{\prime }P\left(G(\alpha )-\Gamma (\alpha )K\right)-P\right]X\left(k_{i}\right)<-X{\left(k_{i}\right)}^{\prime }\left[K^{\prime }R_{w}K+Q\right]X\left(k_{i}\right)<0. \end{array}$$Considering $U\left(k_{i}\right)=-KX\left(k_{i}\right)$ in (Equation 47), we have
(48)$$\begin{array}{c} X{\left(k_{i}\right)}^{\prime }QX\left(k_{i}\right)+U{\left(k_{i}\right)}^{\prime }R_{w}U\left(k_{i}\right)<\Delta V\left(X\left(k_{i}\right)\right). \end{array}$$Assuming that the system described in (Equation 31) is stable in closed-loop, and by applying summation to both sides of (Equation 48) as $k_{i}=0$ to $k_{i}\to \infty$, we obtain,
(49)$$\begin{array}{cc} \; & \sum_{k_{i}=0}^{\infty }{\left[X{\left(k_{i}\right)}^{\prime }QX\left(k_{i}\right)+U{\left(k_{i}\right)}^{\prime }R_{w}U\left(k_{i}\right)\right]}^{J}<X{\left(0\right)}^{\prime }PX\left(0\right),\\ \; & J<X{\left(0\right)}^{\prime }PX\left(0\right),P=M^{-1}. \end{array}$$Then, we applied the Schur complement from Lemma 1 in the inequality (Equation 45) successively, resulting in the LMI provided by Equation (Equation 38). This approach shows that for all admissible *K* and $P=M^{-1}$ resulted by solving the inequality (Equation 38), the inequality (Equation 45) holds. To apply the Schur complement to (Equation 45), one obtains the following result:
(50)$$\begin{array}{cc} \left[\begin{array}{cc} M & G(\alpha )-\Gamma (\alpha )K\\ G{\left(\alpha \right)}^{\prime }-K^{\prime }\Gamma {\left(\alpha \right)}^{\prime } & -K^{\prime }R_{w}K-Q+M^{-1} \end{array}\right] & >0,\\ \sum_{i=1}^{k}\alpha_{i}\left[\begin{array}{cc} M & G_{i}-\Gamma_{i}K\\ G_{i}^{\prime }-K^{\prime }\Gamma_{i}^{\prime } & -K^{\prime }R_{w}K-Q+M^{-1} \end{array}\right] & >0,\\ \left[\begin{array}{cc} M & G_{i}-\Gamma_{i}K\\ G_{i}^{\prime }-K^{\prime }\Gamma_{i}^{\prime } & -K^{\prime }R_{w}K-Q+M^{-1} \end{array}\right] & >0. \end{array}$$Rewriting (Equation 50), one has,
(51)$$\begin{array}{cc} \; & \left[\begin{array}{cc} M & G_{i}-\Gamma_{i}K\\ G_{i}^{\prime }-K^{\prime }\Gamma_{i}^{\prime } & -Q+M^{-1} \end{array}\right]-\left[\begin{array}{c} 0\\ K^{\prime }R_{w} \end{array}\right]R_{w}^{-1}\left[\begin{array}{cc} 0 & R_{w}K \end{array}\right]>0,\\ \; & \left[\begin{array}{ccc} R_{w} & 0 & R_{w}K\\ 0 & M & G_{i}-\Gamma_{i}K\\ K^{\prime }R_{w} & G_{i}^{\prime }-K^{\prime }\Gamma_{i}^{\prime } & -Q+M^{-1} \end{array}\right]>0. \end{array}$$As a result, to avoid inverse matrices inside a convex problem, such as the term $M^{-1}$, both the left-hand and right-hand sides of Equation (Equation 51) can be multiplied by diag(I,I,M). However, bilinearities might appear, so let N=KM such that
(52)$$\begin{array}{c} \left[\begin{array}{ccc} R_{w} & 0 & R_{w}N\\ 0 & M & G_{i}M-\Gamma_{i}N\\ N^{\prime }R_{w} & M^{\prime }G_{i}-N^{\prime }\Gamma_{i}^{\prime } & -MQM+M \end{array}\right]>0. \end{array}$$Rewriting (Equation 52) we obtain,
(53)$$\begin{array}{c} \left[\begin{array}{ccc} R_{w} & 0 & R_{w}N\\ 0 & M & G_{i}M-\Gamma_{i}N\\ N^{\prime }R_{w} & M^{\prime }G_{i}-N^{\prime }\Gamma_{i}^{\prime } & -MQM+M \end{array}\right]+\left[\begin{array}{c} 0\\ 0\\ MQ \end{array}\right]Q^{-1}\left[\begin{array}{ccc} 0 & 0 & QM \end{array}\right]>0. \end{array}$$Applying the Schur complement in (Equation 53) results in the LMI described in (Equation 38). According to Definition 1, which defines a D-stable system with pole allocation, the LMI given in (Equation 39) is used for closed-loop pole allocation in the predictive control system design. In this case, a circumference with a radius ρ and a center at (q,0) was used for the allocation of closed-loop poles in the z plane. The pole placement in specific regions of the *z*-plane using the LMI framework can be described by (Equation 54),
(54)$$\begin{array}{c} \left[\begin{array}{cc} -\rho M & A_{r}M+qM\\ MA_{r}^{\prime }+qQ & -\rho M \end{array}\right]<0. \end{array}$$The LMI (Equation 39) was obtained considering the matrix $A_{r}$ equal to (G(α)−Γ(α)K), which was described in (Equation 42), resulting in (Equation 54) using change of variable: $K=NM^{-1}$. The proof is complete. □

**Theorem** **2.**
*Consider the system (Equation 31) with performance index given by the cost function (Equation 34), if the following optimization problem*

$$\begin{array}{ccc} \underset{\delta ,M,N}{min(\delta )} & \; & \end{array}$$


(55)
$$\begin{array}{cc} s.\;t. & \;\;\;LMI\;(38) \end{array}$$


(56)
$$\begin{array}{cc} \; & \;\;\;LMI\;(39) \end{array}$$


(57)
$$\begin{array}{cc} \; & \left[\begin{array}{cc} M & X(0)\\ X{\left(0\right)}^{\prime } & \delta \end{array}\right]>0, \end{array}$$

*has a feasible solution (δ,M,N) then $U=-NM^{-1}X\left(k_{i}\right)$ is a guaranteed cost control law of system (Equation 31).*


**Proof.** From Theorem 2, the guaranteed cost control $U\left(k_{i}\right)=-NM^{-1}X\left(k_{i}\right)$ is obtained, moreover, from (Equation 57), it follows that $J=X{\left(0\right)}^{\prime }M^{-1}X\left(0\right)\leq \delta$, which means that $U\left(k_{i}\right)=-NM^{-1}X\left(k_{i}\right)$ is a guaranteed cost control law of system (Equation 31). □

Equations (Equation 55)–(Equation 57) form a convex optimization problem with LMI constraints. Therefore, to solve the problem, one can resort to software such as MATLAB (2021b). In Section 5, a comparison was performed between the results of the control system proposed in this work and the results obtained by applying current mode control to the inverter output regulation.

## 5. Simulink Comparison

The proposed LMI-MPC controller was compared with a consolidated power electronics average current mode control (ACMC) scheme [38]. We conducted this comparison using the PWM inverter illustrated in Figure 1 and a resistive load. Table 1 describes the parameters of the inverter.

The adopted ACMC has two control loops, each with a proportional-integral plus resonant controller $\left(P_{I}\left(z\right)+R_{ess}\left(z\right)\right)$. Equation (Equation 58) represents the aforementioned controller whereas Table 2 summarizes the main parameters of such control.
(58)$$\begin{array}{c} P_{I}\left(z\right)=\frac{kz-k_{1}}{z-1}\;and\;R_{ess}\left(z\right)=\frac{k_{2}z-k_{2}}{z^{2}-2z+1}. \end{array}$$

As shown in Table 1, we considered a PWM inverter with parametric uncertainties in the load resistance and the inductance of the filter. Based on the discrete uncertain model of the inverter given in (Equation 14), we can define a polytope of $p=2^{2}$ vertices. The vertices of the polytopic model are: (59)$$\begin{array}{cc} \; & A_{d1}=\left[\begin{array}{cc} 0.991 & 0.237\\ -0.007 & 0.999 \end{array}\right],A_{d2}=\left[\begin{array}{cc} 0.991 & 0.237\\ -0.005 & 0.999 \end{array}\right]\;,A_{d3}=\left[\begin{array}{cc} 0.994 & 0.237\\ -0.007 & 0.999 \end{array}\right]\;, \end{array}$$(60)$$\begin{array}{cc} \; & A_{d4}=\left[\begin{array}{cc} 0.995 & 0.237\\ -0.005 & 0.999 \end{array}\right]\;,B_{d1}=B_{d3}=\left[\begin{array}{c} 0.0008\\ 0.0073 \end{array}\right]\;,B_{d2}=B_{d4}=\left[\begin{array}{c} 0.0006\\ 0.0050 \end{array}\right]\;, \end{array}$$(61)$$\begin{array}{cc} \; & C_{d1}=C_{d2}=C_{d3}=C_{d4}=\left[\begin{array}{cc} 1 & 0 \end{array}\right]. \end{array}$$

The matrices of the discrete uncertain model were obtained from (Equation 3), (Equation 4), and (Equation 10). In the discretization procedure of the uncertain continuous system, we used Zero-order hold as discretization method. Using MATLAB, we applied the c2d command with the matrices of the continuous system and the sampling period as input, and we obtained the discrete matrices in Equations (Equation 59)–(Equation 61).

**Remark** **1.**
*The design of the control system uses Theorem 2, which allows for the insertion of polytopic uncertainties in the resistance and inductance of the inverter output filter. If the converter’s response is not suitable, one can change the position area of the closed-loop poles of the predictive control system by adjusting the radius ρ and the center q of the unit circle in the complex plane z, which changes the region for allocating of the poles.*


The control horizon is selected to be $N_{c}=2$ and the prediction horizon is $N_{p}=3$. Using the vertices of the uncertainty polytope given in (Equation 59)–(Equation 61), we obtain the vertices of the uncertain predictive model with integral action $S_{p}\left(\alpha \right)=\left(G\left(\alpha \right),\;\Gamma \left(\alpha \right),\;F\left(\alpha \right),\;\Phi \left(\alpha \right)\right)$ given in (Equation 31). The LMI region used for pole placement in the design of the predictive control system was a disk with a radius ρ equal to 0.9 and centered at (q,0)=(0,0). The weighting values of $R_{w}$ were chosen to give more weight to the first control law vector $u(k_{i})$ compared to $u(k_{i}+1)$. By using a lower weighting value, we accepted a higher value for $u(k_{i})$ to achieve faster control convergence without a large value in (Equation 34). Define $K={\left[\begin{array}{cc} K_{1} & K_{2} \end{array}\right]}^{\prime }$ and $R_{w}=\left[\begin{array}{cc} R_{w_{1}} & 0\\ 0 & R_{w_{2}} \end{array}\right]$, then (Equation 34) can be rewritten as
(62)$$\begin{array}{c} J=\sum_{k_{i}=0}^{\infty }\left[U{\left(k_{i}\right)}^{\prime }\left(K_{1}^{\prime }R_{w_{1}}K_{1}+K_{2}^{\prime }R_{w_{2}}K_{2}\right)U\left(k_{i}\right)+X{\left(k_{i}\right)}^{\prime }QX\left(k_{i}\right)\right]. \end{array}$$

Hence, the values in the first row of the gain matrix *K* can be designed according to $R_{w_{1}}<<R_{w_{2}}$ to mitigate the second row effect, leading to a potential receding horizon application. In order to increase the purposed $K_{1}>>K_{2}$, the authors employed a trial-and-error fashion to design the weighting values of *Q*. Thus, using the Theorem 2, with known $S_{pi}=\left(G_{i},\;\Gamma_{i},\;F_{i},\;\Phi_{i}\right)$, i=1,2,3,4 and,
(63)$$\begin{array}{c} X\left(0\right)=0_{N_{p}\left(n_{1}+1\right)\times 1},Q=\left[\begin{array}{cc} 100I_{L\times L} & 0_{L\times 2N_{p}}\\ 0_{2N_{p}\times L} & 0.001I_{2N_{p}\times 2N_{p}} \end{array}\right],R_{w}=\left[\begin{array}{cc} 0.1 & 0\\ 0 & 1000 \end{array}\right], \end{array}$$
where $L=n_{1}+1$. The parameters $R_{w}$ and *Q* weight the control law $U(k_{i})$ and the predicted states $X(k_{i})$ in the cost function *J* given in (Equation 34), respectively. Thus, the magnitude of the predictive controller gain (*K*) can be adjusted for FPGA implementation. The LMI-MPC gain matrix *K* is:(64)$$\begin{array}{c} K=\left[\;\begin{array}{ccccc} 205.539 & 148.658 & 35.386 & 1.9\times 10^{-5} & -5.3\times 10^{-5}\\ -3\times 10^{-4} & -2.1\times 10^{-5} & 2.1\times 10^{-6} & 8.7\times 10^{-5} & -6.2\times 10^{-5} \end{array}\right.\\ \left.\begin{array}{cccc} 4\times 10^{-5} & 2.9\times 10^{-5} & -2.7\times 10^{-5} & 4.8\times 10^{-5}\\ -3.6\times 10^{-5} & -3.1\times 10^{-5} & -3.1\times 10^{-5} & -1\times 10^{-4} \end{array}\;\right]. \end{array}$$

Note that in (Equation 64), the designed weighting parameters led to the extreme dominance of the actual augmented states ($x_{e}\left(k_{i}\right)$) and control law ($u(k_{i})$), enabling the receding horizon control principle [39]. Thus, the gain matrix can be written as $K_{e}$: (65)$$\begin{array}{c} K_{e}=\left[\begin{array}{ccc} 205.539 & 148.658 & 35.386 \end{array}\right]. \end{array}$$

**Remark** **2.**
*If the magnitude of the gains for the controller K is too high, leading to computational implementation challenges within the digital control system, Theorem 2 provides a solution by allowing for the adjustment of values for $R_{w}$ and Q. This adjustment aims to achieve controller gains with a magnitude suitable for smooth implementation.*


The ACMC and predictive control applied to the inverter were implemented in MATLAB/Simulink with load resistance equal to R=35
Ω and filter inductance equal to $L_{c}=700$
μF. The simulation to compare the effectiveness of the controller employed the solver ode23tb, and considered a sinusoid reference with 179.6 V amplitude at 60 Hz frequency. In Figure 4 one can observe the performance of both controllers during steady-state full-power operation. The inverter output signal using the ACMC and the proposed LMI-MPC showed practically identical responses.

The aforementioned controllers present almost null steady-state error and, thus, load step change comparison must be done. In this context, the increase of load current of 35% was adopted for the nominal load current, which was applied at 37.5 ms. Through Figure 5, one can verify the outstanding performance of the proposed LMI controller regarding the adopted ACMC for load step disturbance rejection. Figure 4 and Figure 5 illustrate the effectiveness of the LMI-MPC proposed in this work for achieving suitable output tracking and load step disturbance rejection.

Furthermore, it is noteworthy that other research endeavors addressing control systems applied to single-phase inverters, as seen in Ref. [40], also demonstrate satisfactory reference signal tracking. Nevertheless, it should be noted that this control system does not specifically tackle the issue of polytopic uncertainties in the inverter model. In comparison with the control system presented in Ref. [41], the predictive control system proposed in this work presents pole allocation, which is important for a transient adjustment, as well as restriction for inverter initial conditions. These predictive controller tuning parameters enable the inverter to provide a sinusoidal response in accordance with the reference signal, while also effectively addressing transient response under load step disturbances.

In that sense, Section 6 considers the implementation of the LMI controller in FGPA. Table 3 summarizes the results of the comparison between the performance of LMI-MPC and ACMC.

The average current mode control features two control loops with a PI controller plus a resonant controller which work in tandem to correct the steady-state error. This characteristic is reflected in the items: the ripple of errors and MSE, which are described in Table 3. However, the LMI-MPC control demonstrates better performance in regulating the transient response. With proposed predictive control, there is the possibility of pole placement, enabling adjustments to the transient performance of the system’s response. Furthermore, in the proposed predictive control scheme, stability is guaranteed for the inverter despite parametric uncertainties in the load resistance and inductance of the LC filter.

## 6. Hardware-in-the-Loop Simulation

The predictive control system algorithm developed in this work and applied at a power electronics inverter was implemented in VHDL hardware description language. This digital implantation was performed in the development kit Altera DE12-115 Cyclone IV, family EP4CE115F29C7 with 114480 logic elements, and the interface between board and computer was linked by the Joint Test Action Group (JTAG). The VHDL code was built and debugged by using the QUARTUS PRIME LITE EDITION 21.1 along with static real and fixed matrices packages. We considered the PWM inverter illustrated in Figure 1, and during the simulation in FPGA-in-the-loop (FIL) of the predictive control system, we factored in a resistive load.

In the simulation of hardware-in-the-loop of the control system, we used the designed predictive controller $K_{e}$ described in (Equation 65). The inverter was implemented in MATLAB/Simulink, whereas the $K_{e}$ controller was embedded into FPGA. Figure 6 illustrates the implemented proposed control system.

Figure 6 illustrates the FIL block in which the LMI-MPC control was implemented, which can be seen in Figure 7. In the VHDL code, we used 16 bits in the input signals, with 14 bits allocated for representing the fractional part and 1 bit for the integer part (sfixed equal to 1-bit signal). The clock of Altera DE12-115 Cyclone IV kit used in this application was 8.4 Mhz. The “filter_done” flag illustrated in Figure 6 indicates the state machine’s status and marks the end of the sample processing. The “sample_trig_out”flag defines the sample time, featuring aduty cycle of 1 clk (1/40$T_{s}$). The variable xm1 is the voltage attenuated by 100×, xm2 is the inductor current attenuated by 10×, rki is the sinusoidal reference with unitary amplitude, and f = 60 Hz. The PWM pulses that drive the switches are mv1, mv2, mv3, and mv4.

The simulations to evaluate the effectiveness of the controller consider the regulation of the inverter output signal in a sinusoid form with an amplitude of 179.6 V and a frequency of 60 Hz. In this context, we considered 20% load variation about the nominal load, and the disturbance signal was applied at 4.16 ms. Since the nominal load and filter output inductance of the inverter are uncertain, as can be seen in Table 1, the following situations were simulated:(a)$R_{c}$ = 31 Ω and $L_{c}$ = 651 μF.(b)$R_{c}$ = 54 Ω and $L_{c}$ = 886 μF.

In case (a), we considered a load current variation of −20%. Figure 8 illustrates that the voltage output of this converter remained regulated with this external disturbance. The inverter response has proper voltage regulation and robustness with a load variation of −20%, with the mean square error (MSE) between the output and the reference signal equal to 2.33. The transient response of the inverter output with rated load insertion of −20% is illustrated in Figure 9.

We also verified the robustness of the proposed control system by introducing an input step disturbance in the DC bus voltage signal (*E*), which was applied to the inverter at 4.16 ms. Figure 10 shows the inverter response to a disturbance input DC bus of +20 V.

Then, the HIL simulation was performed for case (b), and we considered a load disturbance signal of +20%. Figure 11 illustrates the inverter response.

The MSE between the output and the reference signal in case (b) was 1.868. Figure 8 and Figure 11 demonstrate that the proposed predictive control system tracks the reference sinusoidal signal of the inverter output. Additionally, the voltage output of this converter remained regulated under load variation of −20% in case (a) and +20% in case (b).

**Remark** **3.**
*The closed-loop stability of the control system applied to the inverter is assured by Theorem 2, provided that the load disturbance remains within the range of uncertainties defined in (Equation 10).*


For the application of the predictive control with the resistive-inductive load, we used parameters that represent an AC induction machine of 1 CV, 746 W and a power factor of 0.9. Table 4 presents the parameters for the inverter with a resistive-inductive load.

The vertices of the polytopic to the discrete uncertain model of the inverter given in (Equation 14) with resistive-inductive load are:(66)$$\begin{array}{cc} \; & A_{d1}=\left[\begin{array}{ccc} 0.9991 & 0.2380 & -0.2376\\ -0.0073 & 0.999 & 0.0008\\ 0.0001 & 0.00001 & 0.9968 \end{array}\right]\;,\;A_{d2}=\left[\begin{array}{ccc} 0.9991 & 0.2380 & -0.2373\\ -0.0073 & 0.9991 & 0.0008\\ 0.0001 & 0.00001 & 0.9941 \end{array}\right]\;,\\ \; & A_{d3}=\left[\begin{array}{ccc} 0.9993 & 0.2380 & -0.2376\\ -0.0050 & 0.9994 & 0.0006\\ 0.0001 & 0.00001 & 0.9968 \end{array}\right],\;A_{d4}=\left[\begin{array}{ccc} 0.9993 & 0.2380 & -0.2373\\ -0.0050 & 0.9994 & 0.0005\\ 0.0001 & 0.00001 & 0.9941 \end{array}\right]\;,\\ \; & B_{d1}=B_{d3}=\left[\begin{array}{c} 0.0008\\ 0.0073\\ 0 \end{array}\right]\;,\;B_{d2}=B_{d4}=\left[\begin{array}{c} 0.0006\\ 0.0050\\ 0 \end{array}\right]\;,C_{d1}=C_{d2}=C_{d3}=C_{d4}=\left[\begin{array}{ccc} 1 & 0 & 0 \end{array}\right]. \end{array}$$

The control horizon is selected to be $N_{c}=2$ and the prediction horizon is $N_{p}=3$. Using the vertices of the uncertainty polytope given in (Equation 66), we obtain the vertices of the uncertain predictive model with integral action $S_{p}\left(\alpha \right)=\left(G\left(\alpha \right),\;\Gamma \left(\alpha \right),\;F\left(\alpha \right),\;\Phi \left(\alpha \right)\right)$ given in (Equation 31). The LMI region used in the pole placement for the design of the predictive control system was a disk with a radius ρ equal to 0.6 and centered at (q,0)=(0,0). Using Theorem 2, with known $S_{pi}=\left(G_{i},\;\Gamma_{i},\;F_{i},\;\Phi_{i}\right)$, i=1,2,3,4 and,
(67)$$\begin{array}{c} X\left(0\right)=0_{N_{p}\left(n_{1}+1\right)\times 1},Q=\left[\begin{array}{cc} 100I_{L\times L} & 0_{L\times 2(N_{p}+1)}\\ 0_{2(N_{p}+1)\times L} & 0.001I_{2\left(N_{p}+1\right)\times 2\left(N_{p}+1\right)} \end{array}\right],R_{w}=\left[\begin{array}{cc} 0.1 & 0\\ 0 & 1000 \end{array}\right]. \end{array}$$

Using the receding horizon control principle [39], the designed predictive controller was,
(68)$$\begin{array}{c} K=\left[\begin{array}{cccc} 600.018 & 232.556 & -231.853 & 191.166 \end{array}\right], \end{array}$$
where $K_{x}=\left[\begin{array}{ccc} 600.018 & 232.556 & -231.853 \end{array}\right]$ and $K_{y}=191.166$, and this controllers were described in Figure 2.

In the HIL simulation, we considered $R_{c}$ = 50 Ω and $L_{c1}$ = 800 μH. At 20.8 ms of the simulation, we introduced an external disturbance signal that increased the current of the resistive-inductive load by +20%. Figure 12 illustrates the inverter’s response.

The MSE between the output and the reference signal of the results illustrated in Figure 12 was equal to 0.5197. The predictive control system demonstrated effective regulation for the inverter output signal feeding a resistive-inductive load. Furthermore, following the RL load disturbance, the control system response was robust while maintaining the design specified voltage level.

## 7. Discussion

This manuscript presented an application of predictive control applied to a single-phase PWM inverter with a LC filter. The proposed controller ensures closed-loop stability and regulation of the inverter output voltage. The design of the controller takes into account parametric uncertainty in the filter inductance and the converter’s output load resistance.

We used Theorem 2 in order to obtain the predictive controller applied to the inverter. In this context, the range for the uncertain parameters is defined in state-space models described in (Equation 3) and (Equation 8). Furthermore, we chose suitable values for the variables $R_{w}$ and *Q* in the controller design. $R_{w}$ weights the value of the control law in calculating the cost function *J*, whereas *Q* weights the magnitude of the states of the system. Theorem 2 also enables us to consider a lower bound for the cost function *J* by selecting an initial value for X(0).

For the design of the LMI-MPC controller, the value of $N_{c}$ and $N_{p}$ is defined by the designer. In situations in which these parameters have large magnitudes, the computational cost for designing the controller increases.

We compare the results of the proposed control system and the average current mode control. Figure 5 shows that the transient response of the converter with predictive control under the load disturbance signal presented a shorter settling time and smaller oscillations compared to the inverter response with the ACMC. The LMI-MPC controller uses the state-space feedback with integral action. The response of the control system showed good results for attenuating the effects of step disturbance in the output of the converter. However, it may not effectively mitigate the impact of exogenous signals in the performance of the inverter when encountering disturbances in other frequency ranges, such as high-frequency disturbances.

The sample time used in the FPGA-in-the-loop simulation of the predictive control system was 210 kHz. We performed the FIL for two scenarios: case (a) and case (b). In the first case, we introduced a disturbance signal of –20% in the rated load current of the inverter. Figure 8 illustrates that the control system presented a suitable disturbance rejection and effective sinusoidal-reference tracking. In case (a), we also simulated the predictive control system for disturbance rejection in the DC bus. Figure 10 presents the inverter output with a +20% disturbance in the DC bus voltage (*E*), demonstrating that the proposed control system can adequately regulate the sinusoidal voltage output of the converter.

In case (b), we introduced a disturbance signal in the rated load current of the inverter of +20%. Figure 11 shows that the control system presented a suitable disturbance rejection and effective sinusoidal-reference tracking. The work also presented a FIL simulation of the RL load applied to the inverter. In this context, we used parameters that represent an AC induction machine of 1 CV, 746 W and power factor of 0.9. Figure 12 shows the inverter output with rated resistive-inductive load insertion of +20% and showed suitable tracking of the sinusoidal reference, as well as rejection of the disturbance signal at the output of the system.

## 8. Conclusions

This work proposes a methodology for designing a predictive control system applied to single-phase PWM inverters. It takes into account parametric uncertainties in filter inductance and inverter load resistance, which enhances the robustness of the control design for practical applications. The controller design was formulated in terms of LMIs, and in this context a region was incorporated for allocation of closed-loop poles, which allows the regulation of the inverter output with specific performance criteria. The designed predictive controller is based on state feedback and integral action, allowing reference sine signal tracking and rejection to external load variation disturbances. The results of the control system were obtained by real-time FPGA hardware-in-the-loop simulation. The implementation of the control system in FPGA is promising because, for higher values $N_{p}$, a higher computational cost will be required to determine the control law applied to the inverter. The simulation results of the predictive control system applied to the inverter demonstrated good performance with reference sinusoidal tracking—even with the presence of parametric uncertainties and load disturbance. The mean square error between the output and the reference signal of the results obtained presented adequate magnitudes, which can be observed by tracking the reference signal, even considering polytopic uncertainties in the inverter model. The predictive controller design for the RL load presented the MSE with magnitude mentor among the results obtained. Going forward, our research will focus on developing a predictive control system, applied to single-phase inverter, consisting of two controllers working together: one to disturbance rejection and another for reference signal tracking.

## Figures and Tables

**Figure 1 sensors-24-04325-f001:**
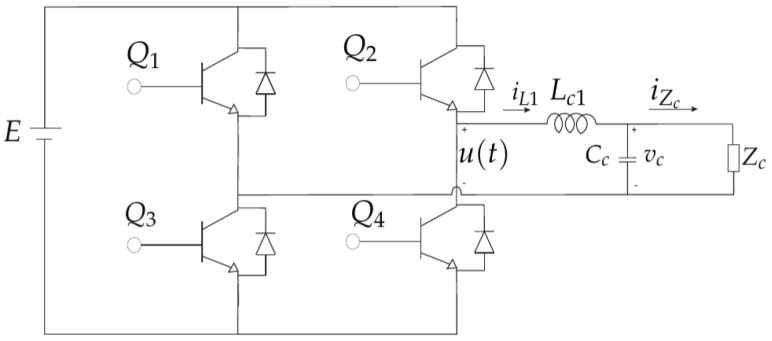
Full-bridge PWM inverter with resistive or resistive-inductive $Z_{c}$ load.

**Figure 2 sensors-24-04325-f002:**
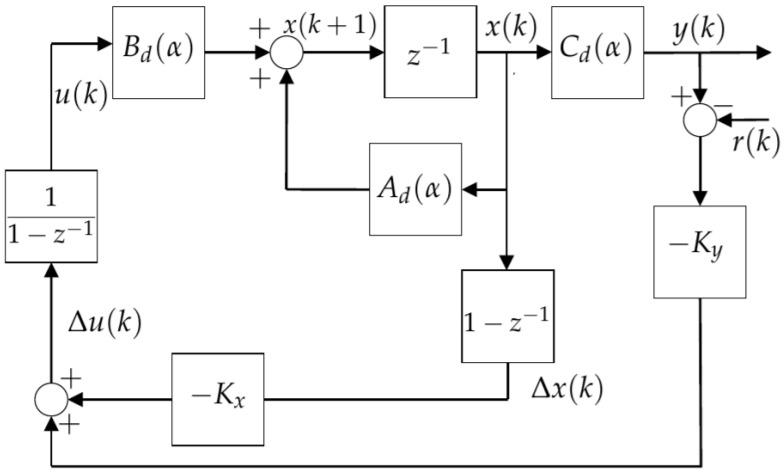
Predictive controller block diagram.

**Figure 3 sensors-24-04325-f003:**
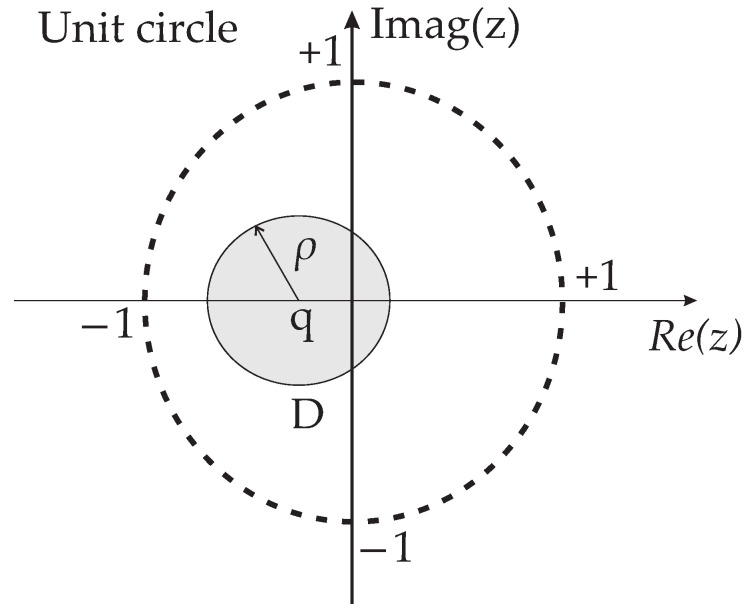
Circular region D for pole placement of closed-loop in complex z-plane.

**Figure 4 sensors-24-04325-f004:**
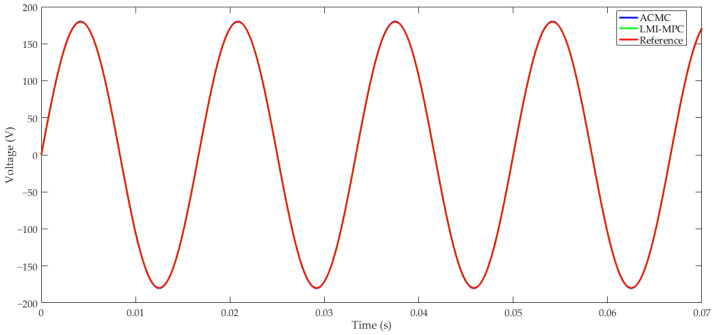
Steady-state in nominal power.

**Figure 5 sensors-24-04325-f005:**
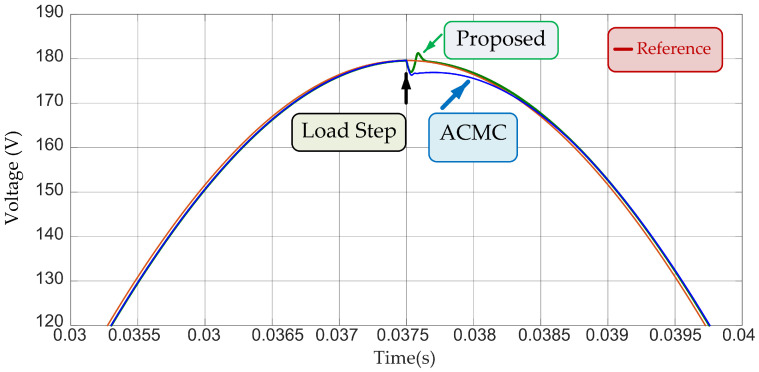
ACMC performance comparison with predictive control.

**Figure 6 sensors-24-04325-f006:**
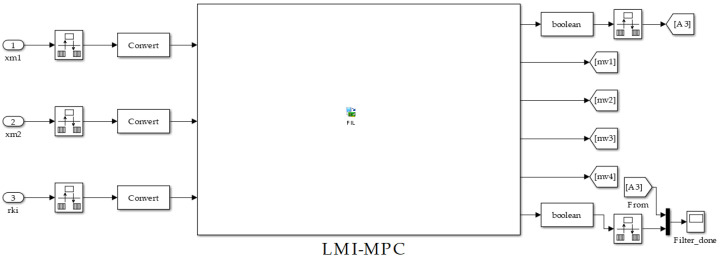
Predictive controller *K* implementation.

**Figure 7 sensors-24-04325-f007:**
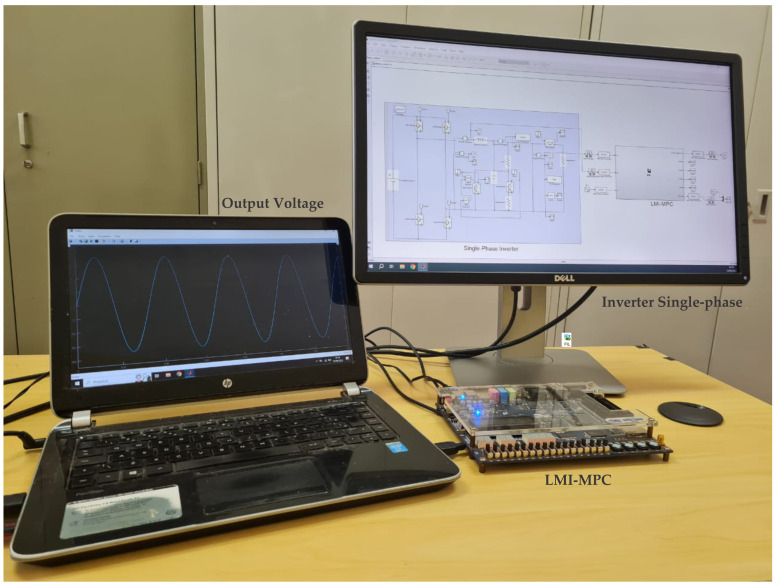
Hardware implementation of the LMI-MPC.

**Figure 8 sensors-24-04325-f008:**
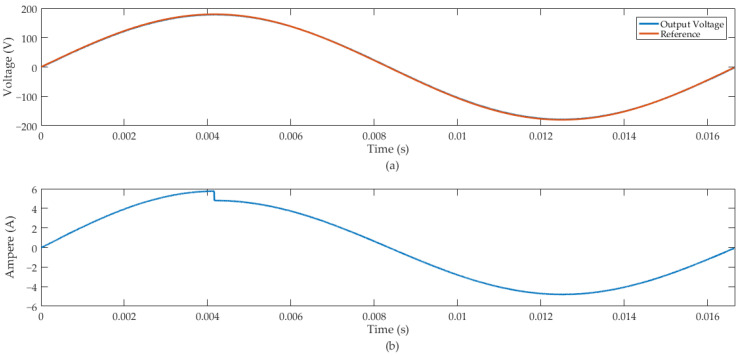
Inverter output with rated load insertion of −20%: (**a**) inverter output voltage and reference signal. (**b**) Inverter output current.

**Figure 9 sensors-24-04325-f009:**
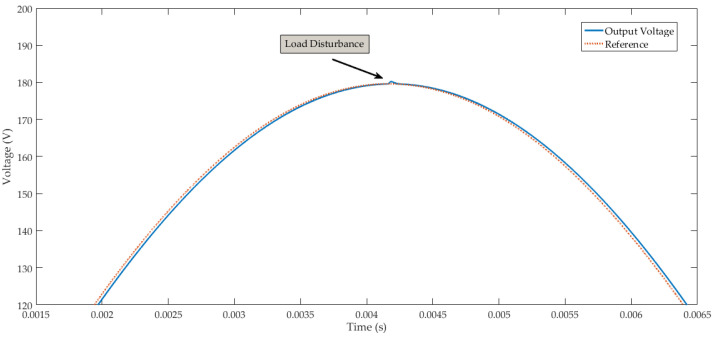
Transient response of the voltage output of the inverter.

**Figure 10 sensors-24-04325-f010:**
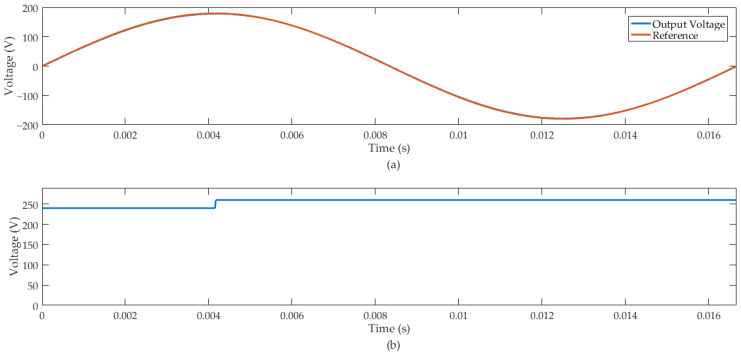
Inverter output with insertion of +20 V in *E*: (**a**) inverter output voltage and reference signal. (**b**) DC bus.

**Figure 11 sensors-24-04325-f011:**
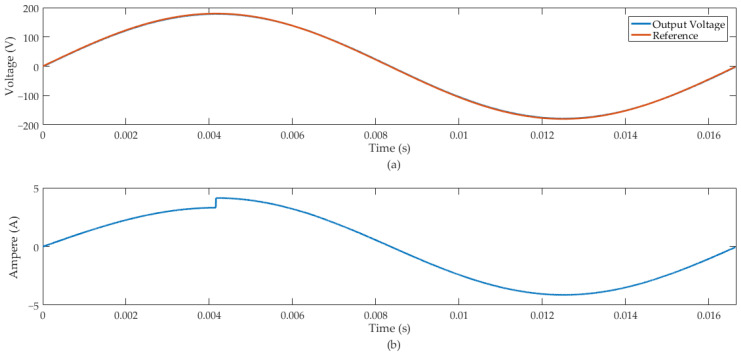
Inverter output with rated load insertion of +20%: (**a**) inverter output voltage and reference signal. (**b**) Inverter output current.

**Figure 12 sensors-24-04325-f012:**
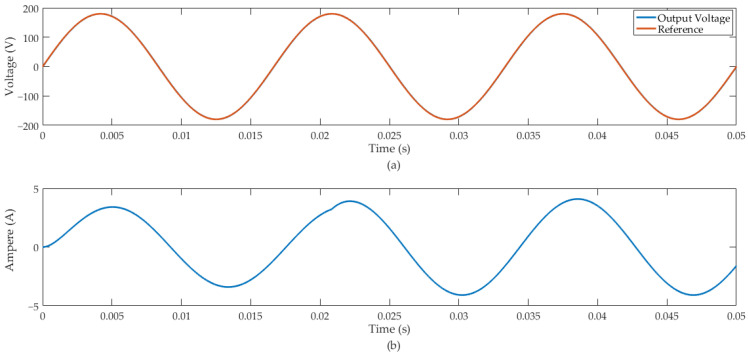
Inverter output with rated resistive-inductive load insertion of +20%: (**a**) inverter output voltage and reference signal. (**b**) Inverter output current.

**Table 1 sensors-24-04325-t001:** PWM inverter parameters with resistive load.

Parameters	Symbol	Value
Load resistance	$R_{c}$	30 Ω < $R_{c}$ < 55 Ω
Filter inductance	$L_{c1}$	650 μH < $L_{c1}$ < 950 μH
Filter capacitance	$C_{c}$	20 μF
DC bus voltage	*E*	240 V
Distribution network RMS voltage	$V_{RMS}$	127 V
Sampling time	$T_{s}$	$\frac{1}{f_{s}}$
Switching frequency	$f_{s}$	210 kHz

**Table 2 sensors-24-04325-t002:** ACMC main compensator parameters.

Voltage Loop	
*k*	0.75
$k_{1}$	0.7418
$k_{2}$	0.007143
Crossing over (kHz)	6
Phase margin (degrees)	85
Current Loop	
*k*	90
$k_{1}$	86.4
$k_{2}$	0.009524
Crossing over (kHz)	20
Phase margin (degrees)	68

**Table 3 sensors-24-04325-t003:** Performance comparison between LMI-MPC and ACMC.

Parameters	LMI-MPC	ACMC
Overshooting (%)	0.89	-
Undershooting (%)	1.4	1.84
Settling time (ms)	0.14	0.64
Ripple of errors	3.89	3.01
MSE	1.83	0.72

**Table 4 sensors-24-04325-t004:** PWM inverter parameters with resistive-inductive load.

Parameters	Symbol	Value
Load resistance	$R_{c}$	$30\;\Omega <R_{c}<55\;\Omega$
Load inductance	$L_{c2}$	45 mH
Filter inductance	$L_{c1}$	$650\;\mu H<L_{c1}<950\;\mu H$
Filter capacitance	$C_{c}$	20 μF
DC bus voltage	*E*	240 V
Grid RMS voltage	$V_{RMS}$	127 V
Sampling time	$T_{s}$	$\frac{1}{f_{s}}$ ms
Switching frequency	$f_{s}$	210 kHz

## Data Availability

Data is contained within the article.

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
