# Peer review of "LMI-Based MPC Design Applied to the Single-Phase PWM Inverter with LC Filter under Uncertain Parameters"

_sensors, 2024, doi:10.3390/s24134325_

Round 1
Reviewer 1 Report
The authors need to consider the following suggestions:
1. The introduction needs to be improved by discussing current articles because only a few new articles have been included in it.
2. It is mandatory for the authors to compare their results with other proposals reported in the literature in order to validate or demonstrate the importance of their contribution to the subject.
3. The authors only test the proposal with an analytical model, I would like to see the performance of the proposal under real conditions (experimental case).
4. The authors need to justify the selection of the parameters used in their simulation. For example, how were selected these values?
5. The conclusion needs to be improved, adding quantitative results not only qualitative results. In addition, it is important to mention, what is next with the investigation.
Author Response
Author's Reply to the Review Report (Reviewer 01)
All authors would like to thank Reviewer 01 for your thorough review of the paper. Your insights and suggestions have been extremely valuable.
Review Report (Reviewer 1)
The authors need to consider the following suggestions:
1. The introduction needs to be improved by discussing current articles because only a few new articles have been included in it.
R: Thank you for your suggestion. In the introduction section, we have incorporated recent articles that discuss the application of predictive control in power electronics, as recognized by the academic community. Additionally, we have included works that describe the utilization of FPGA in control system implementation. The changes are located on page 2 of the manuscript, where they have been marked in red color.
2. It is mandatory for the authors to compare their results with other proposals reported in the literature in order to validate or demonstrate the importance of their contribution to the subject.
R: Thank you for your thoughtful suggestion. We described the control system proposed in this work in the form of LMIs and inserted restrictions for pole allocation, Equation (44) from the manuscript, and polytopic uncertainties in the inverter model. In this context, we inserted information highlighted in red on pages 13 and 14 to describe the contribution of the predictive controller presented in this work to the research area in control systems applied to power electronics, more specifically in single-phase PWM inverters.
3. The authors only test the proposal with an analytical model, I would like to see the performance of the proposal under real conditions (experimental case).
R: The research work that resulted in the writing of this manuscript began with the design of the predictive controller via LMI in MATLAB. After this period, the single-phase inverter was simulated in the Simulink environment.
In a next stage of the studies, the simulation of the controller was carried out using Simscape from MATLAB. In this procedure, the inverter was implemented with the electronic components present in the software library. Then, with the simulation results from the previous steps that showed good performance of the control system, the MPC was embedded in an FPGA, but with the simulation still in MATLAB, which validated the results obtained previously.
Finally, we simulated the control system with the FPGA-embedded MPC, with only the inverter implemented in MATLAB, and this latter procedure is referred to as FPGA-in-the-loop (FIL) in the manuscript.
The simulation via FIL can be observed in several works such as:
de Andrade, N.D.; Godoy, R.B.; Batista, E.A.; de Brito, M.A.G.; Soares, R.L.R. Embedded FPGA Controllers for Current Compensation Based on Modern Power Theories. Energies 2022, 15, 6284. https://doi.org/10.3390/en15176284
Almabrok, A.; Psarakis, M.; Dounis, A. Fast Tuning of the PID Controller in An HVAC System Using the Big Bang–Big Crunch Algorithm and FPGA Technology. Algorithms 2018, 11, 146. https://doi.org/10.3390/a11100146
H. S. Khan and A. Y. Memon, "FIL simulation of sliding mode controller for DC/DC boost converter," 2016 13th International Bhurban Conference on Applied Sciences and Technology (IBCAST), Islamabad, Pakistan, 2016, pp. 112-117, doi: 10.1109/IBCAST.2016.7429864.
The authors emphasize that the experimental case under real conditions is important, and this item would be exposed in suggestion number 5. In a future study, restrictions for control input (duty cycle) will be inserted, as well as analyzing the performance in non-linear loads to observe the performance of the LMI-MPC system under real conditions.
4. The authors need to justify the selection of the parameters used in their simulation. For example, how were selected these values?
R: The parameters used to tune the predictive controller in this work are:
- X(0): Lower bound for the cost function J;
- Rw: weighting for the control law U in the cost function J;
- Q: weighting for the predicted states X(ki) in the cost function J.
You can change the magnitude of the K controller by adjusting these parameters. There are situations in which the solution provides a K with high magnitude, which does not allow the implementation in FPGA. In this case, the value of Rw and Q can be reduced to reduce the magnitude of the designed K controller.
An alternative approach to adjusting the value of K involves manipulating the closed-loop pole allocation region in the control system design. In this context, the methodology proposed in this work enables the selection of a region within the complex z-plane, specifically within the unit circle, for the placement of the closed-loop poles.
This region is bounded by a circle with center q and radius ρ. When considering that the center q is located at the origin, increasing the value of q (0 < q < 1) in the project allows for achieving a slower response, necessitating a K controller with smaller magnitudes. Conversely, reducing the value of ρ results in a faster closed-loop system response, which results in a K controller with large magnitudes, making its implementation in FPGA difficult.
We have included a paragraph in the text, which can be found on page 13 in red, to justify the selection of parameters used in the design.
5. The conclusion needs to be improved, adding quantitative results not only qualitative results. In addition, it is important to mention, what is next with the investigation.
R: Thank you for your suggestion. We have updated the Conclusion section to include quantitative results and future directions stemming from this work. These revisions can be found in red on page 19.
We want to express our sincere gratitude for the valuable suggestion provided by you as a reviewer of our work in the prestigious Electronic Sensors. Your observations and insights have been invaluable in enhancing the quality of our paper. We are committed to carefully consider your suggestions and incorporate them to the best of our ability. Once again, thank you for your significant contribution to our work and the advancement of research in the field.
Reviewer 2 Report
This work proposes a design methodology for predictive control applied to the single-phase PWM inverter. The paper is well organized, however, some issues should be considered to improve it.
1/ The English should be polished, especially, the use of the past tense.
2/ The control scheme should be well introduced, not as a remark,
3/ The discretization of the model should be also considered.
4/Please check all notations (like Od in eq 22).
5/ What does it mean F(alpha) must be quadratic?
6/ The term reliable is introduced with no meaning in this study.
7/ The proof of the theorem is not well conducted to show the guaranteed cost stability of the system. The author can refer to
Robust reliable guaranteed cost piecewise fuzzy control for discrete-time nonlinear systems with time-varying delay and actuator failures
Robust H ∞ output feedback control with pole placement constraints for uncertain discrete-time fuzzy systems
8/Lemma 2 needs a reference.
9/ Some photos should be added to show the hardware implementation.
10/ More remarks should be added in the simulation part to highlight the proposed approach.
Can be improved.
Author Response
Review Report (Reviewer 2)
This work proposes a design methodology for predictive control applied to the single-phase PWM inverter. The paper is well organized, however, some issues should be considered to improve it.
1/ The English should be polished, especially, the use of the past tense.
R: Thank you for your valuable feedback and suggestions on the writing. We have conducted a comprehensive review and revision of the content, with changes highlighted in blue.
2/ The control scheme should be well introduced, not as a remark,
R: Thank you for this review suggestion. Figure 2 was introduced on page 4, with the changes highlighted in red.
3/ The discretization of the model should be also considered.
R: Thank you for these considerations about the discretization procedure of the single-phase inverter continuous system. On page 12 this information was inserted, which is highlighted in red.
4/Please check all notations (like Od in eq 22).
R: We appreciate your suggested correction. The Od notation in equation (22) was indeed incorrect. We have since corrected the Od notation, and it is now marked in red on page 6. Furthermore, we conducted a comprehensive review of the notations used throughout the entire document.
5/ What does it mean F(alpha) must be quadratic?
R: Thank you for your important observation. Only G(alpha) needs to be square, while F(alpha) has a different dimension.
For Np=3, with matrices Ae(alpha) of size 3x3 and Be(alpha) and Ce(alpha) given in (22), the matrix Fi has a size of 3x3. However, when we obtain the predicted system, as can be seen in equation (31) in the paper, we have the following multiplication:
F(alpha)X(ki) (I)
The vector X(ki) represents the predicted states, and since it is a SISO system, the size of X(ki) is 9 x 1. Therefore, for the multiplication given in (I), we properly fill the matrix F with zeros.
We have corrected this information in the text, which is highlighted in red on page 7.
6/ The term reliable is introduced with no meaning in this study.
R: Thank you for this review suggestion. A reliable guaranteed cost control in control theory and engineering is often associated with systems that have actuator or sensor failures. Therefore, the term reliable has been removed from the text. These changes in the text can be seen in red on pages 8, 9, 11.
7/ The proof of the theorem is not well conducted to show the guaranteed cost stability of the system. The author can refer to
Robust reliable guaranteed cost piecewise fuzzy control for discrete-time nonlinear systems with time-varying delay and actuator failures
Robust H ∞ output feedback control with pole placement constraints for uncertain discrete-time fuzzy systems
R: We appreciate your suggestion. In order to prove the theorem, we utilized Definition 1 along with Lemmas 1 and 2. To enhance the proof of Theorem 1, we reworked Lemma 1 and incorporated Definition 1 into the text. These modifications are highlighted in brown text on pages 8, 9, and 10.
8/Lemma 2 needs a reference.
R: Thank you for this review suggestion. Lemma 2 provides an analysis of the Schur Complement. In the updated text, we have incorporated a reference to this item.
9/ Some photos should be added to show the hardware implementation.
R: Thanks for this suggestion. The authors have included a photo depicting the implementation of the LMI-MPC control system on an FPGA connected to the single-phase inverter implemented in MATLAB. This specific implementation is commonly referred to as FPGA-in-the-loop (FIL). The textual modification related to this suggestion can be found on page 15 by referring to Figure 5.
10/ More remarks should be added in the simulation part to highlight the proposed approach.
R: We appreciate your recommendation. Remarks 1, 2, and 3 have been included on pages 13, 14, and 18 of the text, respectively.
We want to express our sincere gratitude for the valuable suggestion provided by you as a reviewer of our work in the prestigious Electronic Sensors. Your observations and insights have been invaluable in enhancing the quality of our paper. We are committed to carefully consider your suggestions and incorporate them to the best of our ability. Once again, thank you for your significant contribution to our work and the advancement of research in the field.
Reviewer 3 Report
I have the following observations:
1. This paper deal with LMI-based MPC Design Applied to the Single-phase PWM Inverter with LC Filter under Uncertain Parameters
2. I suggest you give more references from last 5 years
3. The paper it’s well written
4. I suggest that in (55) use the Tustin discretization instead rectangular discretization and use the paper “Comparative Study of Discrete PI and PR Controls for Single-Phase UPS Inverter MOHAMMAD PARVEZ 1, MOHAMAD FATHI MOHAMAD ELIAS 2, (Member, IEEE), NASRUDIN ABD RAHIM 2,3, (Senior Member, IEEE), FREDE BLAABJERG 4, (Fellow, IEEE), DEREK ABBOTT 1, (Fellow, IEEE), AND SAID F. AL-SARAWI 1, (Member, IEEE) Digital Object Identifier 10.1109/ACCESS.2020.2964603” instead old references [29] and [30] even that ACMC could be remain for comparison.
5. I suggest to summarize the result of comparison In a table and the terms should be : overshooting, ripple of errors, stationary errors.
Author Response
Author's Reply to the Review Report (Reviewer 03)
All authors would like to thank Reviewer 03 for your thorough review of the paper. Your insights and suggestions have been extremely valuable.
Review Report (Reviewer 3)
I have the following observations:
1. This paper deal with LMI-based MPC Design Applied to the Single-phase PWM Inverter with LC Filter under Uncertain Parameters
2. I suggest you give more references from last 5 years
R: Thank you for this review suggestion. We have updated the introduction section with more recent references, and you can find the revised text highlighted in red on page 2.
3. The paper it’s well written
R: Thank you for this consideration of the work.
4. I suggest that in (55) use the Tustin discretization instead rectangular discretization and use the paper “Comparative Study of Discrete PI and PR Controls for Single-Phase UPS Inverter MOHAMMAD PARVEZ 1, MOHAMAD FATHI MOHAMAD ELIAS 2, (Member, IEEE), NASRUDIN ABD RAHIM 2,3, (Senior Member, IEEE), FREDE BLAABJERG 4, (Fellow, IEEE), DEREK ABBOTT 1, (Fellow, IEEE), AND SAID F. AL-SARAWI 1, (Member, IEEE) Digital Object Identifier 10.1109/ACCESS.2020.2964603” instead old references [29] and [30] even that ACMC could be remain for comparison.
R: Thanks for the suggestions. In the text, references [29] and [30] were updated to the most current suggested references. Subsequently, in the process of obtaining (55), we employed the Tustin discretization method. However, the tuning result of the PI controller exhibited a regulated yet oscillatory response. To observe this situation, consider the discrete controllers obtained using the Tustin method,
Voltage Controller
- Phase margin (degrees) = 85
- Crossing over = 6 kHz
PI(z) = ( 0.7571 z - 0.7402)/ z-1 (1)
Current Controller
- Phase margin (degrees) = 70
- Crossing over=19.6 kHz
PI(z) = (89.23 z - 72.02)/(z-1) (2)
The response of the continuous control system designed and discretized via Tustin is shown in Appendix 1 of the document attached, along with the responses to the reviewer's suggestions. In the simulation, a load disturbance of 35% was applied in 0.0357 s.
Therefore, the authors opted for the ZOH discretization method. To design of ACMC , the authors employed the transfer functions of the power electronics converters directly in the Z-domain through the utilization of the Z-transform. Subsequently, we generated the Bode plots using MATLAB's Sisotool. The discrete Proportional-Integral (PI) controller was fine-tuned by observing the desired crossover frequency and phase margin. The choice of the zero for the PI controller was made to minimize the impact of the converter’s pole. Finally, to enhance steady-state accuracy, we incorporated the resonant component in parallel with the PI controller directly within the simulator. This was done to mitigate errors during steady-state operation. The resonant component was tuned to operate at 60Hz, with its equation being discretized using the Zero-Order Hold (ZOH) method.
5. I suggest to summarize the result of comparison In a table and the terms should be : overshooting, ripple of errors, stationary errors.
R: We appreciate your review suggestion. Table 3 has been inserted into the manuscript to summarize the results of the comparison between LMI-MPC and ACMC. This change in the text can be seen in red on pages 13 and 14.
We want to express our sincere gratitude for the valuable suggestion provided by you as a reviewer of our work in the prestigious Electronic Sensors. Your observations and insights have been invaluable in enhancing the quality of our paper. We are committed to carefully consider your suggestions and incorporate them to the best of our ability. Once again, thank you for your significant contribution to our work and the advancement of research in the field.
Round 2
Reviewer 1 Report
The authors have addressed correctly the Reviewer's concerns. Hence, i can recommend accepting the article.
Author Response
The authors would like to express their appreciations to Reviewer 1 for the thorough and thoughtful review of our article, "LMI-based MPC Design Applied to the Single-phase PWM Inverter with LC Filter under Uncertain Parameters". Your experience and constructive feedback were instrumental in refining the content and ensuring the highest quality of the work.
Reviewer 2 Report
I read the revised version. The proof of theorem 1 is not well conducted to show the guaranteed cost D-stability of the system.1/ The authors added a defintion and a lemma to proof it. However, a defintion hasen't a proof, and the lemma is not well stated.
I propose providing the definition of D-stability (pole placement), and incorporating the demonstrations of lemma1 and the definition into the proof of the theorem.
2/ Some related references should be added to update the related references and give some comparisons or explanations.
- Robust reliable guaranteed cost piecewise fuzzy control for discrete-time nonlinear systems with time-varying delay and actuator failures
- Robust H ∞ output feedback control with pole placement constraints for uncertain discrete-time fuzzy systems.
3/ The title of section is not well selected.
4/ The limitation of the method should be discussed.
The quality of English can be improved
Author Response
Author's Reply to the Review Report (Reviewer 02)
All authors would like to express to Reviewer 02 our gratitude for the thorough and careful reviews of the manuscript. Your attention to detail and constructive feedback were extremely valuable. Thank you for your time and effort in helping us improve the submitted work.
Review Report (Reviewer 2)
I read the revised version. The proof of theorem 1 is not well conducted to show the guaranteed cost D-stability of the system.
1/ The authors added a definition and a lemma to proof it. However, a defintion hasen't a proof, and the lemma is not well stated.
I propose providing the definition of D-stability (pole placement), and incorporating the demonstrations of lemma1 and the definition into the proof of the theorem.
R: Thank you for this suggestion on preparing the proof of the Theorem, which addresses guaranteed cost and stability. In this context, we implement your proposal your proposal to define D-Stability, as well as insert into the proof of Theorem 1 the demonstrations of Definition 1 and Lemma 1 from the previous version of the manuscript. The text has been updated on pages 8 through 11, with the changes highlighted in blue.
2/ Some related references should be added to update the related references and give some comparisons or explanations.
- Robust reliable guaranteed cost piecewise fuzzy control for discrete-time nonlinear systems with time-varying delay and actuator failures
- Robust H ∞ output feedback control with pole placement constraints for uncertain discrete-time fuzzy systems.
R: Thank you for this suggestion. The reference titled 'Robust reliable guaranteed cost piecewise fuzzy control for discrete-time nonlinear systems with time-varying delay and actuator failures' was cited in the Introduction section to emphasize the significance of robust guaranteed cost controllers applied to uncertain systems. This modification in the text can be found on page 2, highlighted in blue.
Additionally, the reference titled 'Robust H ∞ output feedback control with pole placement constraints for uncertain discrete-time fuzzy systems' was cited in the definition of the D-Stability system on page 9.
3/ The title of section is not well selected.
R: Thank you for this observation. Upon reviewing the 'Predictive model with integral action' section, it became apparent that it also detailed the design of the LMI-MPC controller. As a result, we have introduced a dedicated section titled 'LMI-MPC Design,' focusing specifically on the LMI-based controller design. In this section, Theorems 1 and 2 are presented in order to design the predictive controller K with integral action. This new section is highlighted in red on page 8.
4/ The limitation of the method should be discussed.
R: Thanks for this suggestion. In the Discussion section, information about limitations of the LMI-MPC controller was inserted and these changes to the text can be seen highlighted in blue on page 20.
The first information is related to the values of Nc and Np. To design the LMI-MPC controller, the value of Nc and Np must be chosen. When these numbers are large, the predictive system matrices are also large, and this increases the computational cost of the project. There are applications in which the predictive controller is designed by the FPGA, and in these situations Nc and Np with high magnitude may not be suitable.
Other Additional information regarding the limitations of the method is associated with the frequency range of the disturbance signal that the controller operates on to mitigate the effects of these exogenous signals on the inverter output. The LMI-MPC controller employs integral action to correct steady-state errors and thereby reject disturbances. In this context, the results of the FIL simulation indicate that the controller satisfactorily attenuates the effects of load disturbance steps. However, it may not achieve the same level of effectiveness for exogenous signals in different frequency bands. To address this, one can observe controllers in the literature that attenuate disturbances in broader frequency ranges, such as the H2 and Hinf controllers.
Updates to the wording of the text are highlighted in green in this version of the text.
The authors would like to express their appreciations to Reviewer 2 for the thorough and thoughtful review of our article, "LMI-based MPC Design Applied to the Single-phase PWM Inverter with LC Filter under Uncertain Parameters", during the second round of reviews. Your experience and constructive feedback were instrumental in refining the content and ensuring the highest quality of the work.